# SURFACE REPRESENTATION IN LiDAR SCENES

## ABSTRACT

Learning from point clouds entails knowledge of local shape geometry. Recent efforts have succeeded in representing synthetic point clouds as surfels. However, these methods struggle to deal with LiDAR point clouds captured from real scans, which are sparse, uneven, and larger-scale. In this paper, we introduce **RealSurf**, a general framework that processes point clouds under extreme conditions like autonomous driving scenarios. We identify several key challenges in applying surface representations to real scans and propose solutions to these challenges: Point Sliding Module that jitters point coordinates within the reconstructed surfels for geometric feature computation, and LiDAR-based surfel reconstruction process that enables models to directly construct surfels from LiDAR point clouds by attenuating unevenness. We evaluate our approach on a diverse set of benchmarks, including nuScenes, SemanticKITTI, and Waymo. RealSurf, with a simple PointNet++ backbone, outperforms its counterparts by a significant margin while remaining efficient. By achieving state-of-the-art results on three benchmarks through a fair and unbiased comparison, RealSurf brings renewed attention to the effectiveness of point-based methods in LiDAR segmentation. Code will be publicly available upon publication.

## 1 INTRODUCTION

Learning from point clouds is a key task in machine learning, with potential applications in many areas including autonomous driving, augmented reality, and robotics. In its most basic incarnation, point cloud learning involves the extraction of geometric features from noisy and sparse points with parameterized models. These geometric features are then consumed by deep models to enable 3D perception tasks. However, many modeling and computational challenges hinder the design of stable and robust deep networks for point cloud processing. First, point clouds are represented by orderless sets, while existing deep learning models such as convolutional neural networks (CNNs) assume inputs with fixed structures. Second, point clouds from LiDAR scenes are extremely sparse and noisy. Therefore, methods developed for synthetic point clouds (Qi et al., 2017a;b; Wang et al., 2019a) do not generalize well to real-world LiDAR scans. Learning robust geometric features from point clouds in the wild remains a challenging and open question.

To address these shortcomings in current methods, we introduce a general framework, termed *RealSurf*, to tackle geometric feature learning from point clouds under extreme conditions. Compared to recent point-based methods (Ran et al., 2022; Ma et al., 2022; Zhao et al., 2021), which target clean and dense points sampled from synthetic shapes, our method is applicable to LiDAR point clouds in outdoor settings. RealSurf tries to answer a central question in learning from point clouds: *how to consistently estimate and leverage geometric features in point cloud networks?* We identify two key challenges faced by previous methods: First, point clouds are sparse, which makes it computationally inefficient and impractical to estimate geometric features from local point groups. Second, estimating consistent normals from point clouds is an ill-defined problem, and learning from globally inconsistent normals is harmful to model performance.

To address these issues, we propose two general approaches that can be potentially adopted by any point-based network. First, we jitter the coordinate of each local point within the reconstructed surfels induced by existing points. Second, some geometric properties can not be consistently estimated. Instead of assuming that these features are consistent during training, we design a novel surfel abstraction to retrieve regular reconstructed triangles by relieving the issue of density variation (unevenness) in LiDAR point clouds.

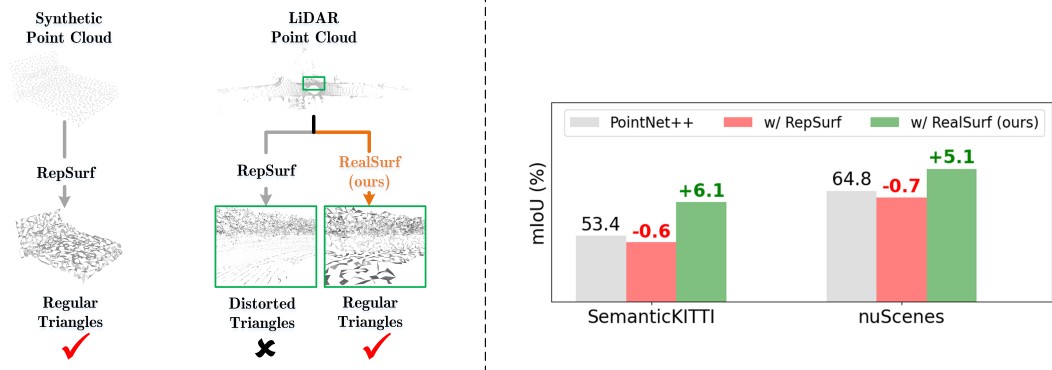

Figure 1: A closer look at our *RealSurf* and RepSurf (Ran et al., 2022). *Left*: Visualization of the reconstructed surface elements, namely *surfels*, on LiDAR point clouds. *Right*: Performance analysis on two popular LiDAR datasets: SemanticKITTI (Behley et al., 2019) and nuScenes (Fong et al., 2022). Here, for fair comparison, we *do not* utilize any techniques mentioned in Sec. 3.4.

We incorporate our proposed modules into a PointNet++ (Qi et al., 2017b) backbone and conduct experiments on a diverse range of datasets including nuScenes, SemanticKITTI, and Waymo. Our results suggest that a simple neural network can achieve state-of-the-art performance on these challenging datasets if equipped with useful geometric features. We hope this finding can motivate future work that combines machine learning and geometry processing.

We summarize our key contributions as follows:

- We introduce a streamlined framework to process LiDAR point clouds. With a simple PointNet++ backbone, it achieves state-of-the-art on several challenging datasets (*i.e.*, SemanticKITTI, nuScenes, and Waymo).
- We analyze challenges in learning geometric features from LiDAR point clouds, addressing a long-standing question in this domain: *why do point-based methods, such as PointNet++, underperform when operating on LiDAR point clouds?*
- We propose solutions to these challenges, namely point sliding module and surfel abstraction. Both designs are general and can be readily adopted by any point-based networks.
- We will open-source our code upon publication to facilitate future development.

## 2  RELATED WORK

### 2.1  POINT CLOUD SEGMENTATION

Previous methods for point cloud segmentation have predominantly relied on four representations: point, 2D projection, voxel, and multi-representation fusion.

**Point-based methods** are commonly used for indoor-scene point clouds characterized by uniform density, a small number of points, and a limited range of scenes. The pioneering work, PointNet (Qi et al., 2017a), utilizes per-point multi-layer perceptrons (MLP) to learn from raw point clouds. Subsequent work has explored various network designs based on graphs (Wang et al., 2019b; Zhou et al., 2021), pseudo grids (Thomas et al., 2019; Li et al., 2018; Xu et al., 2020b; Lai et al., 2023), relations (Liu et al., 2019b; Zhao et al., 2019; Liu et al., 2019a; Zhao et al., 2021; Ran et al., 2021; Xu et al., 2021b; Xiang et al., 2021; Lai et al., 2022) or simply per-point MLP (Qi et al., 2017b; Liu et al., 2020; Ma et al., 2022; Qian et al., 2022). Recently, efforts have been made to address the challenges posed by large-scale LiDAR point clouds. Despite advances such as adaptive sampling and fast random sampling introduced by PointASNL (Yan et al., 2020) and RandLA-Net (Hu et al., 2020), expressive local aggregators like KPConv (Thomas et al., 2019) and BAAFNet (Qiu et al., 2021) remain less competitive than more recent approaches for LiDAR point clouds.

**Projection-based methods** have demonstrated efficiency for LiDAR point clouds by converting 3D point clouds into 2D grids (*i.e.*, range view (Wu et al., 2018; 2019; Xu et al., 2020a; Milioto et al., 2019), bird's-eye view (Zhang et al., 2020; Tatarchenko et al., 2018) or both (Liong et al.,

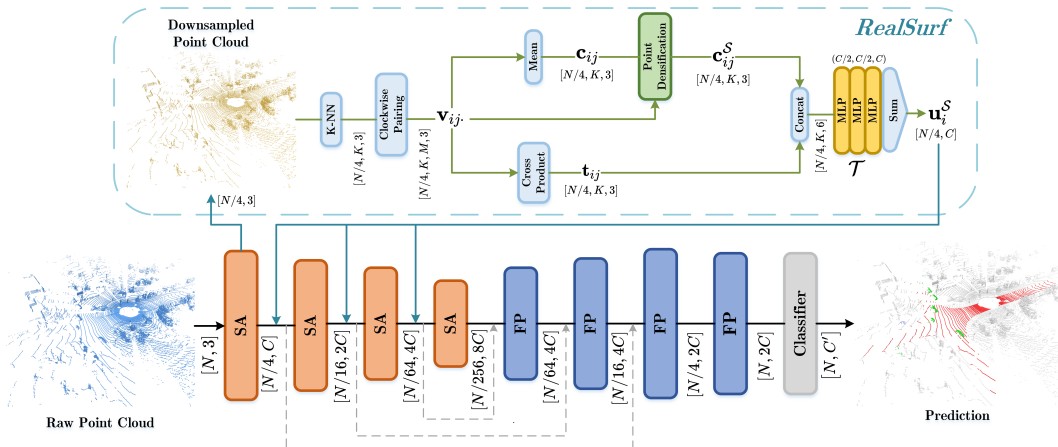

Figure 2: An overview of RealSurf on PointNet++ (Qi et al., 2017b). $N$, $C$ and $C'$ refer to the number of input points, base channels and segmentation label categories, respectively. By default, we set the number of queried neighbors for surfel abstraction to 8, denoted by $K$. $M$ represents the number of vertices of one triangle, and thus $M = 3$. The abbreviations "SA" and "FP" stand for set abstraction and feature propagation in Qi et al. (2017b), respectively.

2020)), thereby enabling the use of 2D convolution. However, the 3D-to-2D projection unavoidably introduces changes in 3D topology and hinders the modeling of complete geometric information, causing inadequate segmentation performance. **Voxel-based methods** (Cheng et al., 2021; Ye et al., 2022) have gained widespread adoption due to their balanced computation costs and performance when integrated with sparse convolutions (Graham et al., 2018). Cylinder3D (Zhu et al., 2021) deforms grid voxels to cylinder-shaped ones and utilizes an asymmetrical network to enhance the performance. **Multi-representation fusion methods** (Xu et al., 2021a; Hou et al., 2022) combine several representations above for segmentation. SPVNAS (Tang et al., 2020) leverages point-voxel representation with per-point MLP and Neural Architecture Search for an efficient architecture.

## 2.2 Surface Representation for Point Clouds

Representing small-scale point clouds as surface elements, or "surfels" (Pfister et al., 2000), has recently gained attention in the community for its successful use in combination with deep learning models. Inspired by triangular and umbrella surfels (Foorginejad & Khalili, 2014), RepSurf (Ran et al., 2022) pre-computes surface features as local geometric information for coordinate input. MaskSurf (Zhang et al., 2022) further incorporates surfels into masked auto-encoding models. While these methods have demonstrated high efficiency and impressive performance, their application to larger-scale LiDAR point clouds, remains challenging and unexplored. This paper aims to analyze the difficulties and propose potential solutions.

## 3 Surface Representation in LiDAR Scenes

In Sec. 3.1, we provide an overview of preliminaries for RealSurf. In Sec. 3.2, we identify three main challenges that arise in computing surface features from point clouds in real-world scenarios. In Sec. 3.3, we present two significant contributions of our proposed method: Point Sliding Module that addresses unevenness and sparsity in LiDAR point clouds, and surfel abstraction that improves the reconstructed surfel quality while addressing unevenness. Finally, in Sec. 3.4, we propose optimized training techniques for point-based networks to compete with voxel-based methods.

### 3.1 Preliminaries

Our method extends RepSurf (Ran et al., 2022), a surface representation that is learned from point clouds. RepSurf (Ran et al., 2022) investigates the tangent plane equation $a(x - x_i) + b(y - y_i) = 0$ and parameterizes the tangent plane as $\mathbf{t}_i = (x_i, y_i, a, b, ax_i + by_i)$, where $(a, b)$ is the normal vector and $(x_i, y_i)$ is the coordinate.

To operate in the 3D space, RepSurf (Ran et al., 2022) introduces a triangle-based representation, Triangular RepSurf, and a multi-surfel representation, Umbrella RepSurf. Triangular RepSurf describes the local geometry of a point $\mathbf{x}_i = (x_i, y_i, z_i)$ by constructing a triangle with the point and its two neighbors (denoted as $\mathbf{x}_i^1$ and $\mathbf{x}_i^2$):

$$\mathbf{p}_i = \frac{1}{3} \left( \mathbf{x}_i + \mathbf{x}_i^1 + \mathbf{x}_i^2 \right), \tag{1}$$

$$\mathbf{t}_i = (a_i, b_i, c_i, a_i x_i + b_i y_i + c_i z_i), \tag{2}$$

where $(a_i, b_i, c_i)$ and $\mathbf{p}_i$ are the normal vector and the centroid cooridnate of the constructed triangle.

However, a single point $\mathbf{x}_i$ is usually contained by multiple surfels. To capture fine-grained geometry of $\mathbf{x}_i$, Umbrella RepSurf uses an umbrella multi-surfel representation (Foorginejad & Khalili, 2014). The geometry around a point $\mathbf{x}_i$ is defined by the aggregation of $K$ adjacent triangular surfels:

$$\mathbf{u}_i = \mathcal{A}\left(\{\mathcal{T}\left([\mathbf{p}_{ij}, \mathbf{t}_{ij}]\right), \forall j \in \{1, \ldots, K\}\}\right), \tag{3}$$

where $\mathbf{p}_{ij}$ and $\mathbf{t}_{ij}$ denote the centroid and Triangular RepSurf features of one of the $K$ adjacent surfels around $\mathbf{x}_i$, respectively; $\mathcal{A}$ and $\mathcal{T}$ are the aggregation module and a learnable transformation function. We refer readers to RepSurf (Ran et al., 2022) for more details. In this paper, we focus on Umbrella RepSurf since it is a more stable and empirically effective representation with a larger receptive field compared to Triangular RepSurf.

## 3.2 CHALLENGES IN REAL SCENES

Although RepSurf (Ran et al., 2022) has demonstrated considerable success in tasks involving dense and uniform point clouds, there are several remaining modeling and computational challenges to adopt RepSurf to real LiDAR scan processing (A typical example of LiDAR point clouds is shown in Figure 3). We include these challenges as follows:

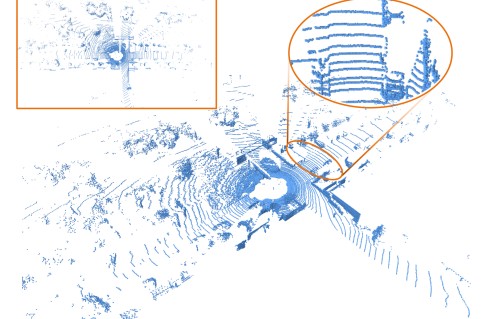

- **Sparsity**. Objects that are farther away appear indistinct due to the sparse points, making it difficult for models to accurately capture their surfel geometry.

- **Density variation**. Density variation affects model expressiveness. For example, a large receptive field is suitable for points farther away but not for near points.

Figure 3: An example of a LiDAR point cloud panorama, along with both its bird's-eye view (top left) and a magnified region near the egocenter (top right).

- **Larger scale**. The considerable number of LiDAR points creates a computational burden for neighbor querying during surfel abstraction, which is not present in synthetic point clouds with fewer points.

As shown in Figure 6, these challenges jointly prevent RepSurf to produce regular and coherent triangles from LiDAR point clouds. The goal of RealSurf is to construct regular triangles by densifying point clouds as presented later.

## 3.3 REALSURF

**Overview.** As shown in Figure 2, RealSurf uses a PointNet++ (Qi et al., 2017b) backbone. To perform surfel abstraction, we firstly downsample point clouds by FPS and feed them into RealSurf. Each point $\boldsymbol{x}_i$ is paired with its $K = 8$ neighbors clockwise to form triangles, which collectively makes up an umbrella surfel. Then, we compute surface features such as centroid coordinates and normals, and use the Point Sliding Module to enhance surface representation. After concatenating centroid coordinates and normals, we use MLPs to map the features into a high-level latent space. Finally, the features are processed by subsequent stages in the backbone.

**Free lunch of augmentation from surfels with Point Sliding module.** To handle sparse point clouds, previous methods (Qi et al., 2017b) simply adopt a learning-based strategy (MLPs followed by max-pooling) to extract features from sparse point groups. However, this does not fundamentally solve this problem, especially for extremely sparse points. To address this challenge, we propose a Point Sliding Module (as shown in Figure 4) to densify irregular and sparse points.

Given a point set $\mathcal{S}$ with $K$ points, the Point Sliding Module jitters the coordinate of a point within its corresponding constructed surfel. The output point through Point Sliding Module is defined as follows:

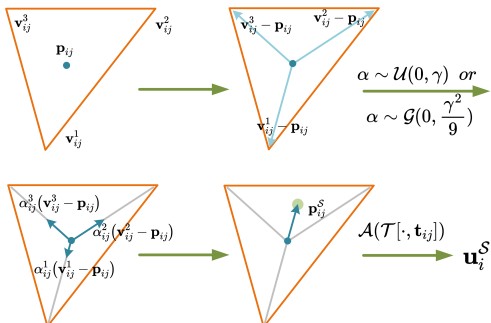

$$\mathbf{p}_{ij}^{\mathcal{S}} = \mathbf{p}_{ij} + \sum_{m=1}^{3} \alpha_{ij}^m (\mathbf{v}_{ij}^m - \mathbf{p}_{ij}), \quad (4)$$

where $\boldsymbol{v}_{ij}^m, m = 1, 2, 3$ are three vertices of the $j$-th triangle and $\boldsymbol{p}_{ij}$ is the centroid; $\alpha_{ij}^m$ is randomly sampled from a distribution (details presented in the next paragraph) to control the corresponding edge vector $\mathbf{v}_{ij}^m - \mathbf{p}_{ij}$. Typically, $\alpha_{ij}^m$ is in the range of $[0, 1]$. We define Umbrella RepSurf through Point Sliding Module as:

Figure 4: Illustration of Point Sliding Module. Given a reconstructed triangle in an umbrella surfel with three vertices $\mathbf{v}_{ij}^m$, $m \in \{1, 2, 3\}$, we can obtain the coordinate of centroid $\mathbf{p}_{ij}$ by averaging the three vertices. Afterwards, we calculate the edge vector $\mathbf{v}_{ij}^m - \mathbf{p}_{ij}$ for each vertex. Then, we sample the scale factor $\alpha$ from a uniform or Gaussian distribution for each edge vector, and compute the offset of the centroid by summation of the randomly scaled edge vectors. This offset allows the centroid to slide on the triangular surfels. Finally, we feed the jittered centroid $\mathbf{p}_{ij}^{\mathcal{S}}$ into the learnable transformation $\mathcal{T}$, followed by the aggregation function $\mathcal{A}$ to compute the surface representation $\mathbf{u}_i^{\mathcal{S}}$.

$$\mathbf{u}_i^{\mathcal{S}} = \mathcal{A}\left(\left\{\mathcal{T}\left(\left[\mathbf{p}_{ij}^{\mathcal{S}}, \mathbf{t}_{ij}\right]\right), \forall j \in \{1, \ldots, K\}\right\}\right). \quad (5)$$

We explore two variants for $\alpha_{ij}^m$: *Uniform* Point Sliding Module and *Gaussian* Point Sliding Module. That is, $\alpha_{ij}^m$ is sampled from a uniform distribution $\alpha_{ij}^m \sim \mathcal{U}(0, \gamma)$ or a Gaussian distribution $\alpha_{ij}^m \sim \mathcal{N}(0, \frac{\gamma^2}{9})$, where $\gamma$ is a hyperparameter to tune for both cases and we also limit $\alpha_{ij}^m$ to $[0, 1]$.

**Downsampling is the key to LiDAR-based surface representation.** LiDAR point clouds are density-varying and contain more points compared to synthetic and indoor point clouds. While previous methods such as Zhu et al. (2021); Tang et al. (2020) use raw point clouds as input due to the unique pattern of LiDAR point clouds, this pattern can severely impact surface representation (as shown in Figure 8) by causing reconstructed triangles to align along circular lines. Furthermore, the large number of input points can negatively affect the efficiency of the model, and there is a trade-off between the number of input points

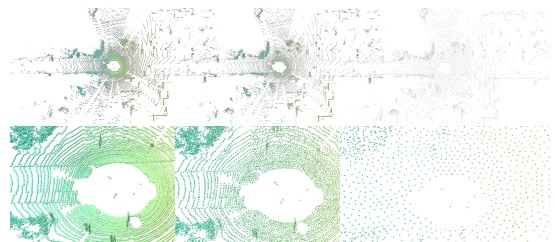

Figure 5: A sequence of LiDAR point clouds with or without downsampling. *Left*: raw point cloud, *Middle*: downsampled (factor: 1/4) point cloud, *Right*: downsampled (factor: 1/16) point cloud.

and the efficiency of surface feature computation. For instance, Waymo (Sun et al., 2020) has around 160,546 input points, while ScanNet (Dai et al., 2017) has around 30,545 input points, and surface features are computed for each point. To address these issues, we propose a surfel abstraction process that allows our method to operate on abstracted surface representations. Specifically, we use farthest point sampling (FPS) to gradually downsample the point clouds and compute RealSurf features along the downsampling process, which are then aggregated into abstract points.

### 3.4 BRIDGING THE GAP BETWEEN POINT-BASED METHODS AND VOXEL-BASED METHODS

Recent voxel-based methods and fusion-based methods dominate LiDAR segmentation, outperforming its point-based counterpart. To revive point-based methods, we examine two critical issues in

| Method | Reference | **mIoU** | **fwIoU** | barrier | bicycle | bus | car | construction | motorcycle | pedestrian | traffic cone | trailer | truck | driveable | other flat | sidewalk | terrain | manmade | vegetation |
|---|---|---|---|---|---|---|---|---|---|---|---|---|---|---|---|---|---|---|---|
| PolarNet (Zhang et al., 2020) | NeurIPS'21 | 69.4 | 87.4 | 72.2 | 16.8 | 77.0 | 86.5 | 51.1 | 69.7 | 64.8 | 54.1 | 69.7 | 63.5 | 96.6 | 67.1 | 77.7 | 72.1 | 87.1 | 84.5 |
| Cylinder3D (Zhu et al., 2021) | CVPR'21 | 77.2 | 89.9 | **82.8** | 29.8 | 84.3 | 89.4 | 63.0 | 79.3 | 77.2 | 73.4 | 84.6 | 69.1 | 97.7 | **70.2** | **80.3** | 75.5 | 90.4 | 87.6 |
| AMVNet (Liong et al., 2020) | IJCAIW'21 | 77.3 | 90.1 | 80.6 | 32.0 | 81.7 | 88.9 | 67.1 | **84.3** | 76.1 | 73.5 | 84.9 | 67.3 | 97.5 | 67.4 | 79.4 | 75.5 | 91.5 | 88.7 |
| SPVNAS (Tang et al., 2020) | ECCV'20 | 77.4 | 89.7 | 80.0 | 30.0 | 91.9 | 90.8 | 64.7 | 79.0 | 75.6 | 70.9 | 81.0 | 74.6 | 97.4 | 69.2 | 80.0 | **76.1** | 89.3 | 87.1 |
| (AF)²-S3Net (Cheng et al., 2021) | CVPR'21 | 78.3 | 88.5 | 78.9 | **52.2** | 89.9 | 84.2 | **77.4** | 74.3 | 77.3 | 72.0 | 83.9 | 73.8 | 97.1 | 66.5 | 77.5 | 74.0 | 87.7 | 86.8 |
| 2D3DNet†∘ (Genova et al., 2021) | ECCV'22 | 80.0 | 90.1 | 83.0 | 59.4 | 88.0 | 85.1 | 63.7 | 84.4 | 82.0 | 76.0 | 84.8 | 71.9 | 96.9 | 67.4 | 79.8 | 76.0 | 92.1 | 89.2 |
| LidarMultiNet†* (Ye et al., 2022) | ArXiv'22 | 81.4 | 90.9 | 80.4 | 48.4 | 94.3 | 90.0 | 71.5 | 87.2 | 85.2 | 80.4 | 86.9 | 74.8 | 97.8 | 67.3 | 80.7 | 76.5 | 92.1 | 89.6 |
| **RealSurf** | **Ours** | 80.1 | 90.9 | 81.4 | 36.8 | **93.2** | **91.8** | 77.2 | 83.4 | **78.9** | 74.8 | **87.3** | **76.2** | 97.7 | 66.2 | 79.9 | 75.5 | **92.6** | 89.3 |

†: Multi-frame as input, ∗: Extra information for training (e.g., 3D detection labels) and two-stage refinement instead of end-to-end, ∘: Both LiDAR and RGB as input.

Table 1: Comparison on the *nuScenes test set* with the metric of mIoU (%). We only report methods published before Sep. 28, 2023 for comparison (refer to the official website for details).

| Mod. | Method | Reference | **mIoU** | road | sidewalk | parking | other-ground | building | car | truck | bicycle | motorcycle | other-vehicle | vegetation | trunk | terrain | person | bicyclist | motorcyclist | fence | pole | traffic sign |
|---|---|---|---|---|---|---|---|---|---|---|---|---|---|---|---|---|---|---|---|---|---|---|
| Point | PointNet++ (Qi et al., 2017b) | NeurIPS'17 | 20.1 | 72.0 | 41.8 | 18.7 | 5.6 | 62.3 | 53.7 | 0.9 | 1.9 | 0.2 | 0.2 | 46.5 | 13.8 | 30.0 | 0.9 | 1.0 | 0.0 | 16.9 | 6.0 | 8.9 |
| Point | TangentConv (Tatarchenko et al., 2018) | CVPR'18 | 40.9 | 83.9 | 63.9 | 33.4 | 15.4 | 83.4 | 90.8 | 15.2 | 2.7 | 16.5 | 12.1 | 79.5 | 49.3 | 58.1 | 23.0 | 28.4 | 8.1 | 49.0 | 35.8 | 28.5 |
| Point | PointASNL (Yan et al., 2020) | CVPR'20 | 46.8 | 87.4 | 74.3 | 24.3 | 1.8 | 83.1 | 87.9 | 39.0 | 0.0 | 25.1 | 29.2 | 84.1 | 52.2 | 70.6 | 34.2 | 57.6 | 0.0 | 43.9 | 57.8 | 36.9 |
| Point | RandLA-Net (Hu et al., 2020) | CVPR'20 | 55.9 | 90.5 | 74.0 | 61.8 | 24.5 | 89.7 | 94.2 | 43.9 | 29.8 | 32.2 | 39.1 | 83.8 | 63.6 | 68.6 | 48.4 | 47.4 | 9.4 | 60.4 | 51.0 | 50.7 |
| Point | KPConv (Thomas et al., 2019) | ICCV'19 | 58.8 | 90.3 | 72.7 | 61.3 | 31.5 | 90.5 | 95.0 | 33.4 | 30.2 | 42.5 | 44.3 | 84.8 | 69.2 | 69.1 | 61.5 | 61.6 | 11.8 | 64.2 | 56.4 | 47.4 |
| Point | **RealSurf** | **Ours** | 70.7 | 91.0 | **77.2** | **69.5** | 34.6 | **92.7** | **97.1** | 56.1 | 62.1 | 59.9 | 59.3 | 86.2 | **75.2** | 71.4 | 77.6 | 48.8 | **69.4** | **66.0** | 71.4 | |
| Voxel | MinkowskiNet (Choy et al., 2019) | CVPR'19 | 53.2 | 88.4 | 71.4 | 57.1 | 22.6 | 90.4 | 94.0 | 27.5 | 26.4 | 24.5 | 18.4 | 83.5 | 65.3 | 65.8 | 40.5 | 46.7 | 13.5 | 62.5 | 54.0 | 59.1 |
| Voxel | PolarNet (Zhang et al., 2020) | CVPR'20 | 54.3 | 90.8 | 74.4 | 61.7 | 21.7 | 90.0 | 93.8 | 22.9 | 40.3 | 30.1 | 28.5 | 84.0 | 65.5 | 67.8 | 43.2 | 40.2 | 5.6 | 61.3 | 51.8 | 57.5 |
| Voxel | (AF)²-S3Net (Cheng et al., 2021) | CVPR'21 | 69.7 | 91.3 | 72.5 | 68.8 | **53.5** | 87.9 | 94.5 | 39.2 | 65.4 | **86.8** | 41.1 | 70.2 | 68.5 | 53.7 | **80.7** | **80.4** | **74.3** | 63.2 | 61.5 | 71.0 |
| V+P | SPVNAS (Tang et al., 2020) | ECCV'20 | 67.0 | 90.2 | 75.4 | 67.6 | 21.8 | 91.6 | 97.2 | **56.6** | 50.6 | 50.4 | 58.0 | 86.1 | 73.4 | 71.0 | 67.4 | 67.1 | 50.3 | 66.9 | 64.3 | 67.3 |
| V+P | Cylinder3D (Zhu et al., 2021) | CVPR'21 | 68.9 | **92.2** | 77.0 | 65.0 | 32.3 | 90.7 | 97.1 | 50.8 | **67.6** | 63.8 | 58.5 | 85.6 | 72.5 | 69.8 | 73.7 | 69.2 | 48.0 | 66.5 | 62.4 | 66.2 |
| V+P | PVKD† (Hou et al., 2022) | CVPR'22 | 71.2 | 91.8 | 77.5 | 70.9 | 41.0 | 92.4 | 97.0 | 53.5 | 67.9 | 69.3 | 60.2 | 86.5 | 73.8 | 71.9 | 75.1 | 73.5 | 50.5 | 69.4 | 64.9 | 65.8 |
| V+R | 2DPASS∘ (Yan et al., 2022) | ECCV'22 | 72.9 | 89.7 | 74.7 | 67.4 | 40.0 | 93.5 | 97.0 | 61.1 | 63.6 | 63.4 | 61.5 | 86.2 | 73.9 | 71.0 | 77.9 | 81.3 | 74.1 | 72.9 | 65.0 | 70.4 |

†: Multi-frame as input, ∘: Both LiDAR and RGB as input.

Table 2: Comparison on the *SemanticKITTI test set* with the metric of mIoU (%). We only report methods published before Sep. 28, 2023 for comparison (refer to the official website for details). V = Voxel, P = Point, R = RGB, Mod. = Modality.

current pipelines: class balance and positive/negative balance. Our final method leads to competitive performance in LiDAR segmentation. In addition, these techniques can potentially benefit any future point-based methods.

**Class Balancing** Class imbalance is a major issue in LiDAR segmentation, where the foreground points are far less than the background points. Voxel-based methods tackle this problem by manual ground-truth augmentation (Yan et al., 2018), which requires an instance database and hyperparameter tuning. Contrarily, RealSurf handles it with a simple point cloud mixing technique, which improves the diversity of foreground points. Given two point clouds $\mathcal{X}$ and $\mathcal{Y}$, we concatenate them and produce a new point cloud $\mathcal{Z}$. Then, our model takes $\mathcal{Z}$ as input and makes per-point predictions.

**Positive/Negative Sample Balancing** Voxel-based methods widely adopt Lovász-Softmax loss (Berman et al., 2018) to improve performance by assigning appropriate weights to small objects and false negatives. Our empirical results show that it does not effectively improve point-based methods. To balance positive and negative samples, we instead use online hard example mining (OHEM) (Shrivastava et al., 2016). In our experiments, we set the threshold to 0.7 and adjust the minimal ratio of kept samples according to the proportion of foreground points in each dataset.

# 4 EXPERIMENTS

We evaluate RealSurf on three commonly used datasets for LiDAR-based segmentation: nuScenes (Fong et al., 2022), SemanticKITTI (Behley et al., 2019), and Waymo Open Dataset (Sun et al., 2020). We additionally perform ablation studies to verify the impact and efficacy of each design choice of RealSurf.

## 4.1 IMPLEMENTATION

**Training Details** We use the AdamW (Loshchilov & Hutter, 2017) optimizer with an initial learning rate of $3 \times 10^{-3}$, a weight decay of $0.01$, and a scheduler that linearly warms up the learning rate for 1500 iterations and then linearly decreases it. All models are trained on 8 GPUs. The per-GPU batch size is respectively 2, 1, and 1 for the nuScenes, SemanticKITTI, and Waymo datasets. We

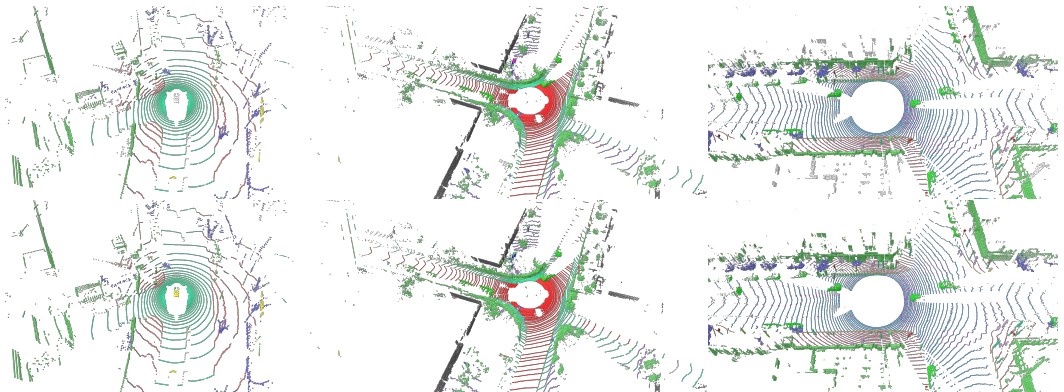

Figure 6: Visualization of ground-truth labels (top row) and the prediction results with our RealSurf (bottom row) on larger-scale LiDAR point cloud datasets from left to right. *Left*: nuScenes (Fong et al., 2022), *Middle*: SemanticKITTI (Behley et al., 2019), *Right*: Waymo (Sun et al., 2020).

| Method | mIoU | car | truck | bus | other-vehicle | motorcyclist | bicyclist | pedestrian | sign | traffic-light | pole | cone | bicycle | motorcycle | building | vegetation | tree-trunk | curb | road | lane-marker | other-ground | walkable | sidewalk |
|---|---|---|---|---|---|---|---|---|---|---|---|---|---|---|---|---|---|---|---|---|---|---|---|
| RGBV-RP Net (Xu et al., 2021a) | 62.6 | 94.8 | 67.4 | 74.9 | 33.0 | 0.0 | 77.3 | 88.6 | 68.0 | 28.6 | 74.7 | 37.6 | 53.8 | 64.9 | 96.5 | 86.4 | 67.1 | 70.9 | 90.9 | 23.7 | 24.0 | 68.8 | 84.4 |
| CAVPercep (Zhu et al., 2021) | 63.7 | 93.6 | 62.8 | 68.1 | 22.8 | **1.8** | 71.7 | 87.3 | 67.0 | 29.4 | 74.2 | 55.0 | 55.6 | 60.8 | 96.1 | 86.4 | 68.0 | 68.3 | 91.2 | 41.2 | 46.1 | 70.5 | 83.1 |
| SPVCNN++§ (Tang et al., 2020) | **67.7** | **95.1** | 67.7 | 75.6 | **35.1** | 0.0 | **85.1** | **91.5** | **73.2** | **31.9** | 78.8 | **61.4** | 65.9 | **73.6** | 90.8 | 86.5 | 70.5 | **75.5** | 91.7 | 41.3 | 40.2 | 71.5 | 85.8 |
| SegNet3DV2° (Li et al., 2022) | 70.4 | 95.7 | 69.0 | 79.7 | 37.0 | 0.0 | 88.7 | 92.6 | 71.8 | 30.0 | 80.8 | 65.9 | 69.5 | 76.9 | 97.1 | 88.1 | 72.7 | 76.4 | 93.2 | 49.4 | 52.6 | 75.4 | 87.2 |
| LidarMultiNet†* (Ye et al., 2022) | 71.1 | 95.8 | 70.5 | 81.4 | 35.4 | 0.7 | 90.6 | 93.2 | 73.8 | 33.1 | 81.3 | 64.6 | 69.9 | 76.7 | 97.3 | 88.9 | 73.5 | 76.5 | 93.2 | 50.3 | 53.8 | 75.4 | 87.9 |
| **RealSurf (ours)** | **67.7** | **95.1** | **68.4** | **80.6** | 32.2 | 0.0 | 78.9 | 90.6 | 70.1 | 28.3 | **79.0** | 52.1 | **67.7** | 72.7 | **97.0** | **87.1** | 71.1 | 74.3 | **92.7** | **44.0** | **48.9** | **71.8** | **85.9** |

†: Multi-frame as input, ∗: Extra information for training (e.g., 3D detection labels) and two-stage refinement instead of end-to-end, ○: Both LiDAR and RGB as input, §: Variant of SPVNAS (Tang et al., 2020) that consumes more computation and apply a longer training schedule.

Table 3: Comparison on the *Waymo test set* with the metric of mIoU (%). We only report officially qualified methods published before Sep. 28, 2023 for comparison (refer to the official website for details).

adopt random flipping, rotation, and point cloud mixing (Nekrasov et al., 2021) to augment data. We additionally use the OHEM loss (Shrivastava et al., 2016) and Fade Strategy (Wang et al., 2021).

**Evaluation Details** We use mIoU for all experiments (and fwIoU–frequency-weighted IoU on nuScenes). For a fair comparison on validation sets, we do not adopt any techniques (*e.g.*, TTA, model ensemble) to further boost the performance. For testing on public servers, we average the predictions of 12 runs for each submission. In contrast to other leading entries on these benchmarks, we *do not* use model ensemble, multi-frame inputs, RGB information, or additional annotations (*e.g.*, detection labels). Our results in the test set are produced through a model trained on singe frames without any of the aforementioned techniques.

## 4.2 NUSCENES

**Dataset** nuScenes (Fong et al., 2022) contains 40,000 frames from 1000 scenes, which are captured by a 32-beam LiDAR sensor in a duration of 20 seconds and sampled at 20Hz. We use the official training/validation splits provided by nuScenes. For LiDAR segmentation, each point is categorized into one of 16 semantic labels. Each frame has ∼34,720 points.

**Results** In Table 1, we evaluate RealSurf on the nuScenes test set. Among all published methods that are not trained with extra information (*e.g.*RGB information in Genova et al. (2021), detection labels in Ye et al. (2022)) or multi-frame input, RealSurf achieves state-of-the-art performance on this benchmark. In particular, for the first time a point-based method achieves better performance than previous state-of-the-art voxel-based methods. Notably, RealSurf outperforms many popular voxel-based methods by a large margin: +1.7% over $(AF)^2$-S3Net (Cheng et al., 2021), +2.7% over SPVNAS (Tang et al., 2020), +2.9% over Cylinder3D (Zhu et al., 2021). Furthermore, our method even outperforms 2D3DNet (Genova et al., 2021) (this method uses RGB inputs), which further validates the performance of our method.

| Modality | Method | Sem.KITTI | nuScenes | #Params |
|---|---|---|---|---|
| Point | KPConv[†] (Thomas et al., 2019) | 54.1 | 66.8 | 14.9M |
| | Point Trans.[†] (Zhao et al., 2021) | 53.7 | 64.7 | 7.8M |
| | PointNet++ | 51.3 | 63.6 | 1.0M |
| | **RealSurf** | 62.3 | 71.4 | 1.0M |
| | PointNet++ (×2) | 53.5 | 66.7 | 3.9M |
| | **RealSurf** (×2) | 68.5 | 75.4 | 3.9M |
| | PointNet++ (×4) | 55.1 | 67.3 | 15.3M |
| | **RealSurf** (×4) | **70.2** | **77.7** | 15.3M |
| Voxel | Cylinder3D (Zhu et al., 2021) | 65.9 | 76.1 | 55.8M |
| | LidarMultiNet[‡] (Ye et al., 2022) | 69.1 | 76.1 | 29.2M |

Table 4: Comparison on *SemanticKITTI & nuScenes val set* with the metric of mIoU (%). We also evaluate our RealSurf in different scales of model capacity (×1, ×2, ×4) with its counterparts vanilla PointNet++. †: Our implementation if the official results unavailable. ‡: Our implementation if the official code is not available.

| RealSurf | | Class Balance | | | P/N Balance | | mIoU | Δ |
|---|---|---|---|---|---|---|---|---|
| rec. | slide | mix | mix3d | gt | ohem | lovasz | | |
| *baseline* | | | | | | | 53.4 | – |
| *RealSurf (ours)* | | | | | | | **62.3** | **+8.9** |
| ✓ | | | | | | | 57.9 | +4.5 |
| ✓ | ✓ | | | | | | **59.5** | **+6.1** |
| ✓ | ✓ | | | ✓ | | | 60.8 | +1.3 |
| ✓ | ✓ | | ✓ | | | | 60.6 | +1.1 |
| ✓ | ✓ | ✓ | | | | | 61.7 | +2.2 |
| ✓ | ✓ | ✓ | | | | ✓ | 59.8 | -1.9 |
| ✓ | ✓ | ✓ | | | ✓ | | **62.3** | **+0.5** |

Table 5: Ablation study of the design of RealSurf and our point-based pipeline on *SemanticKITTI val set* with the evaluation metric of mIoU (%). "Δ" means the improvement compared to the baseline or the best result in previous stage.

## 4.3 SEMANTICKITTI

**Dataset**  Unlike nuScenes dataset, each frame of SemanticKITTI (Behley et al., 2019) is captured by a Velodyne-HDLE64 LiDAR in a larger range, and thus has more instances and exhibits more complex patterns. SemanticKITTI is an extension of the KITTI Visual Odometry dataset (Geiger et al., 2012), which consists of 22 stereo sequences. SemanticKITTI provides per-point labels from 19 categories and splits the 22 sequences into training set (Sequence 00-10 except 08, count: 19,130 frames), validation set (Sequence 08, count: 4,071) and testing set (Sequence 11-21, count: 20,351). Each frame contains ∼121,415 points.

**Results**  In Table 2 we compare RealSurf to all existing segmentation methods on the SemanticKITTI benchmark. For a fair comparison, we only consider published methods that do not leverage multi-frame inputs nor RGB information. Among all existing point-based methods, RealSurf outperforms the previous state-of-the-art method KPConv (Thomas et al., 2019) by 11.9% mIoU. RealSurf even outperforms popular voxel-based architectures, such as $(AF)^2$-S3Net (Cheng et al., 2021) by 1.0% mIoU and Cylinder3D (Zhu et al., 2021) by 1.8% mIoU.

## 4.4 WAYMO

**Dataset**  Waymo Semantic Segmentation dataset (V1.3.2) (Sun et al., 2020) contains 23,691; 5,976; and 2,982 frames in the training set, validation set, and test set, respectively. Each point is classified into 23 semantic categories. Different from the datasets we have mentioned before, Waymo provides larger-scale (∼160,546 points) and more diverse (sampled from 1,100 sequences at a higher frequency) LiDAR scenes.

**Results**  RealSurf achieves 67.7% mIoU on the Waymo test set (Table 3), the same performance as SPVCNN++ (a variant of SPVNAS (Tang et al., 2020)). In addition, our method outperforms Cylinder3D's (Zhu et al., 2021) variant CAVPercep by 4.0%, and outperforms RGB V-RPNet, an RPVNet (Xu et al., 2021a) variant (which is based on LiDAR and RGB fusion), by 5.1%.

## 4.5 RESULTS ON VALIDATION SET

For a closer look at the comparison of RealSurf to other popular methods, we provide empirical results on SemanticKITTI and nuScenes validation set in Table 4. Here, we train LidarMultiNet (Ye et al., 2022) on these validation sets with segmentation labels only for fair comparison with other methods. The performance of RealSurf significantly improves when we increase the dimension of the feature space. For example, RealSurf achieved +6.2% / +4% relative improvement when doubling the feature dimension on SemanticKITTI / nuScenes, which are both significantly better than PointNet++. In addition, RealSurf is able to outperform all voxel-based and point-based methods with significant margin (+1.1% and +1.6% against the best ones on SemanticKITTI and nuScenes).

## 4.6 ABLATION STUDY

In this section, we perform additional ablation studies on the design choices of RealSurf to verify the efficacy of each component. For fast evaluation, we use smaller PointNet++ backbone, termed as RealSurf ($\times$1), for the following experiments.

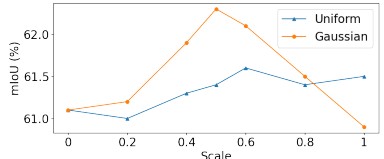

**Point Sliding Module.** As shown in Figure 7, Gaussian Point Sliding Module with the scale of 0.5 performs the best. In addition, we find that Gaussian performs better than Uniform in most scales. We conjecture that Point Sliding Module benefits from Gaussian distribution that move points in a relatively small range. Thus, RealSurf can learn from surface representation with different randomly augmented centroids but still focus on the centroid of real vertices.

Figure 7: Ablation study of Point Sliding Module.

**LiDAR-based Surfel Abstraction.** We explore different stages of SA to perform surfel abstraction in the following without using Point Sliding Module. As shown in the Table 6, the utilization of RealSurf during the surfel abstraction phase of the second stage of SA yields a noteworthy 4.5% improvement in comparison to the RepSurf default surfel abstraction, which actually decreases performance by 0.6%. This observation serves to highlight the efficacy of our

| Recons. | none | 0 | 1 | 2 | 3 |
|---------|------|------|------|------|------|
| mIoU | 53.4 | 52.8 | **57.9** | 56.4 | 53.8 |

Table 6: Performing surfel reconstruction (recons.) after $i$ times of downsampling (*i.e.*, FPS). none: w/o surface representation

LiDAR-based surfel abstraction approach. It is our contention that performing surfel abstraction after FPS affords RealSurf the opportunity to assimilate a more comprehensive range of geometric details, while simultaneously effectively addressing the issue of unevenness inherent in raw LiDAR point clouds. Conversely, the model degeneration from the default surfel abstraction may be attributed to the suboptimal surfel quality. Therefore, determining the appropriate stage for surfel abstraction remains a hyperparameter that requires careful consideration. Our findings indicate that the second stage achieves an optimal balance between efficiency and performance.

**Overall Architecture.** Shown in Table 5, we ablate the design of our method's architecture in terms of RealSurf modules, Class Balancing, and Positive/Negative Sample Balancing. The baseline is PointNet++ with common augmentation methods (*i.e.*, random rotation, random flipping). Empirically, the combination of point cloud mixing (mix), OHEM loss (ohem), and a full usage of RealSurf modules produce the best performance. First of all, the surfel abstraction process and the Point Sliding Module in RealSurf can progressively boost the performance by 4.5% and 1.6%. Notably, as mentioned in Sec. 3.4 that point cloud mixing can alleviate class imbalance, our empirical results further confirm this argument with the improvement of 2.2%. At the same time, OHEM loss improves the performance by 0.5%.

## 5 CONCLUSION & LIMITATION

We present RealSurf, a streamlined approach to point cloud learning in the wild. RealSurf achieves state-of-the-art performance on a diverse set of challenging benchmarks without using complex model ensemble or multi-frame testing. For the first time, we show that a simple point-based method can outperform voxel-based methods on point cloud segmentation. RealSurf only leverages a PointNet++ backbone, orders of magnitude simpler than prior works. Our results also suggest several venues for future development. First, point-based methods do not scale up to deal with extremely large point clouds. Exploring efficient architectures for point clouds will generally benefit point-based methods. Second, the RealSurf features used in this paper can be further used as self-supervised pre-training signals, in addition to their usage in supervised training. Finally, combining RealSurf with voxel-based methods will lead to better hybrid approaches. We hope RealSurf can become a standard model for point-based point cloud processing and motivate a rethinking of the point cloud representation.

## ETHICS STATEMENT

In our paper, we strictly adhere to ICLR ethical research standards and laws. All datasets we use are publicly available, and all relevant publications and source codes are appropriately cited.

## REPRODUCIBILITY STATEMENT

We adhere to ICLR reproducibility standards and ensure the reproducibility of our work in some ways as follows:

- We provide the codes of our main experiments in the supplementary material (code.zip), which includes the pretrained models and some demo samples.
- Detailed framework and more experiments are presented in the Appendix.

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

| Method | w/o noise | | | | | 10% noise | | | | |
|---|---|---|---|---|---|---|---|---|---|---|
| | 16 | 32 | 64 | 128 | 256 | 16 | 32 | 64 | 128 | 256 |
| PointNet++ | 56.32 | 58.40 | 64.19 | 69.40 | 74.50 | 51.91 | 55.31 | 58.40 | 66.79 | 73.59 |
| baseline | 60.96 | 68.88 | 74.91 | 78.83 | 79.91 | 59.06 | 66.11 | 72.24 | 76.06 | 79.08 |
| w/ Gaussian | 61.35 | 69.54 | 75.05 | 78.04 | 79.25 | **59.61** | 66.76 | 71.89 | 75.68 | 78.52 |
| w/ Uniform | **61.83** | **69.88** | **75.64** | **78.94** | **81.58** | 59.30 | **66.86** | **73.21** | **76.79** | **79.24** |

Table 7: Robustness of training on extremely sparse point clouds with or without noises from ScanObjectNN to simulate real-world point clouds. baseline: surface representation without Point Densification Module, Gaussian: our Gaussian Point Densification Module, Uniform: our Uniform Point Densification Module.

---

**Algorithm 1** PyTorch-Style Pseudocode of Sectorized FPS

```
# xyz: coordinates of a point set
# num_sector: number of sectors
angle = atan2(xyz[..., 0], xyz[..., 1])
sector_range = linspace(angle.min(), angle.max(), num_sector + 1)

# sectorized fps
new_xyz = []
for idx in range(num_sector):
    selected_idx = where((angle >= sector_range[s]) & (angle < sector_range[s + 1]))
    new_sector_xyz = farthest_point_sampling(xyz[selected_idx])
    new_xyz.append(new_sector_xyz)

out = cat(new_xyz, 0)
return out
```

---

## A  MODEL DETAILS

### A.1  ROBUSTNESS

Sparsity and noises are common in LiDAR point clouds. To evaluate the robustness of surface representation to sparse and noisy point clouds during training, we conduct several experiments on ScanObjectNN Uy et al. (2019), a real-scene object classification dataset. We use overall accuracy (%) as the evalution metric. For extreme settings, we use 16, 32, 64, 128, and 256 points instead. Further, to evaluate model robustness against noises, we add 10% Gaussian noises to each point cloud. That is, we randomly select 10% the points from each point cloud and replace them with Gaussian noise.

**Robustness to sparsity.** As shown in Table 7, we observe that RealSurf can greatly improve the robustness of its backbone PointNet++ Qi et al. (2017b) by a large margin ($\sim$10%). At the same time, our Point Densification Module can further enhance its robustness by $\sim$1%. We conjecture this is because our Point Densification Module can densify point clouds implicitly, and thus lead to a more robust model. Overall, Uniform Point Densification Module works better than Gaussian Point Densification Module in this setting.

**Robustness to noises.** As shown in Table 7, the surface representation can improve model robustness in the 10% noise setting. Compared to baselines, it introduces a performance gain by 7%$\sim$14%. In this challenging case, our Point Densification Module can still improve the robustness of surface representation by 0.1%$\sim$1.5%.

### A.2  CLASS BALANCING

Class imbalance can be vital for the training of point-based methods on LiDAR segmentation. To handle this problem, we adopt point cloud mixing inspired by Nekrasov et al. (2021). Given two point clouds $\mathcal{S}_1$ and $\mathcal{S}_2$, the output $\mathcal{S}_{final}$ of the mixed point cloud is as follows:

$$\mathcal{S}_{final} = [Augment(\mathcal{S}_1), Augment(\mathcal{S}_2)], \tag{6}$$

where $[\cdot, \cdot]$ means the operation of concatenation, and $Augment$ means augmentation, i.e., Random Rotation (z-axis aligned, range: $[-\frac{\pi}{4}, \frac{\pi}{4}]$) $\rightarrow$ Random Flip (prob: 0.5). Different from Nekrasov

| Method | mIoU (%) | #Params (M) | Inference (s/sample) | Memory (GB) |
|---|---|---|---|---|
| Cylinder3D (Zhu et al., 2021) | 65.9 | 55.8 | 0.13 | 1.62 |
| LiDARMultiNet (Ye et al., 2022) | 69.1 | 29.2 | 0.41 | 6.35 |
| *RealSurf* (×2) | 68.5 | 3.9 | 1.16 | 2.06 |
| *RealSurf* (×4) | 70.2 | 15.3 | 1.72 | 4.14 |

Table 8: Overhead analysis.

et al. (2021), we do not apply random sub-sampling, random scaling, or random rotation along the other axes.

### A.3 POSITIVE/NEGATIVE SAMPLE BALANCING

As mentioned in the paper, point-based method usually requires positive/negative sample balancing. To that end, we adopt online hard example mining (OHEM) Shrivastava et al. (2016) in our pipeline. If the probability of the predicted class is lower than a threshold, we think we need to keep this sample for learning. In addition, this requires a minimum ratio to keep samples for learning. That is, if the probabilities of the predicted class for most samples are above the threshold, we need to keep that ratio of samples for learning. We set the threshold to 0.7. In order to obtain the hyperparameter of minimal kept ratio, we set it as twice the ratio of foreground points in each dataset. That is, 0.01, 0.005, and 0.001 for SemanticKITTI Behley et al. (2019), nuScenes Fong et al. (2022), and Waymo Sun et al. (2020), respectively.

### A.4 EFFICIENT FPS

The low efficiency of FPS makes point-based methods less competitive compared to voxel-based methods. To alleviate this problem, we propose Sectorized FPS to speed up FPS. Sectorized FPS saves 30%~40% training time with almost no performance loss. A PyTorch-style Pseudocode of the implementation of Sectorized FPS is shown in Algorithm 1. As shown in Figure 8, we provide an example to show the difference between the results of vanilla FPS and those of Sectorized FPS. To balance the performance and efficiency, we set the hyperparameter of the number of sectors to 12 in all experiments. Besides, we perform Sectorized FPS only in the first and second stage. For other stages, we perform vanilla FPS instead. We do not apply Sectorized FPS during inference.

## B OVERHEAD ANALYSIS

As shown in Table 8, we provide comparison of overhead among different methods in terms of the number of parameters, inference speed (s per sample), and memory cost.

## C VISUALIZATION

As shown in Figure 9, 10 and 11, we provide additional visualizations of the predictions by RealSurf compared with the ground-truth labels on the dataset of nuScenes Fong et al. (2022), SemanticKITTI Behley et al. (2019), and Waymo Sun et al. (2020), respectively.

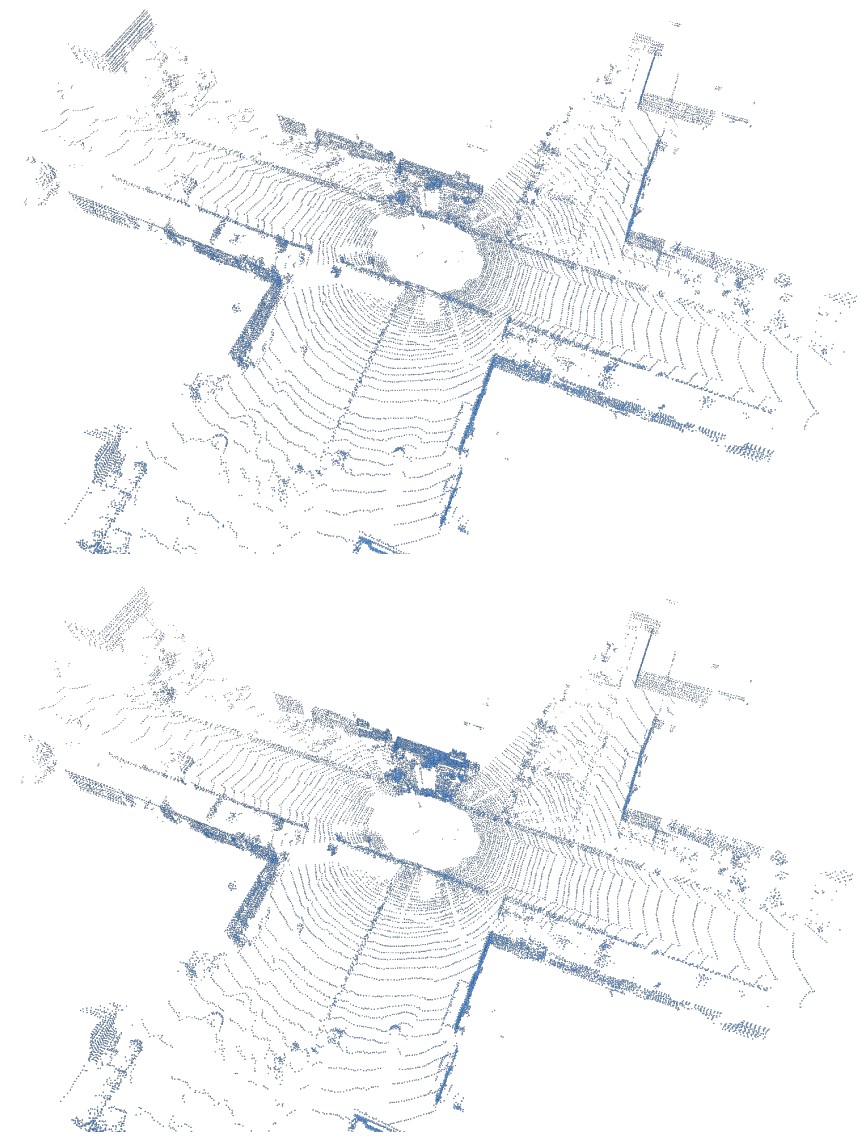

Figure 8: Comparison between vanilla FPS (top) and Sectorized FPS (bottom) in the first stage.

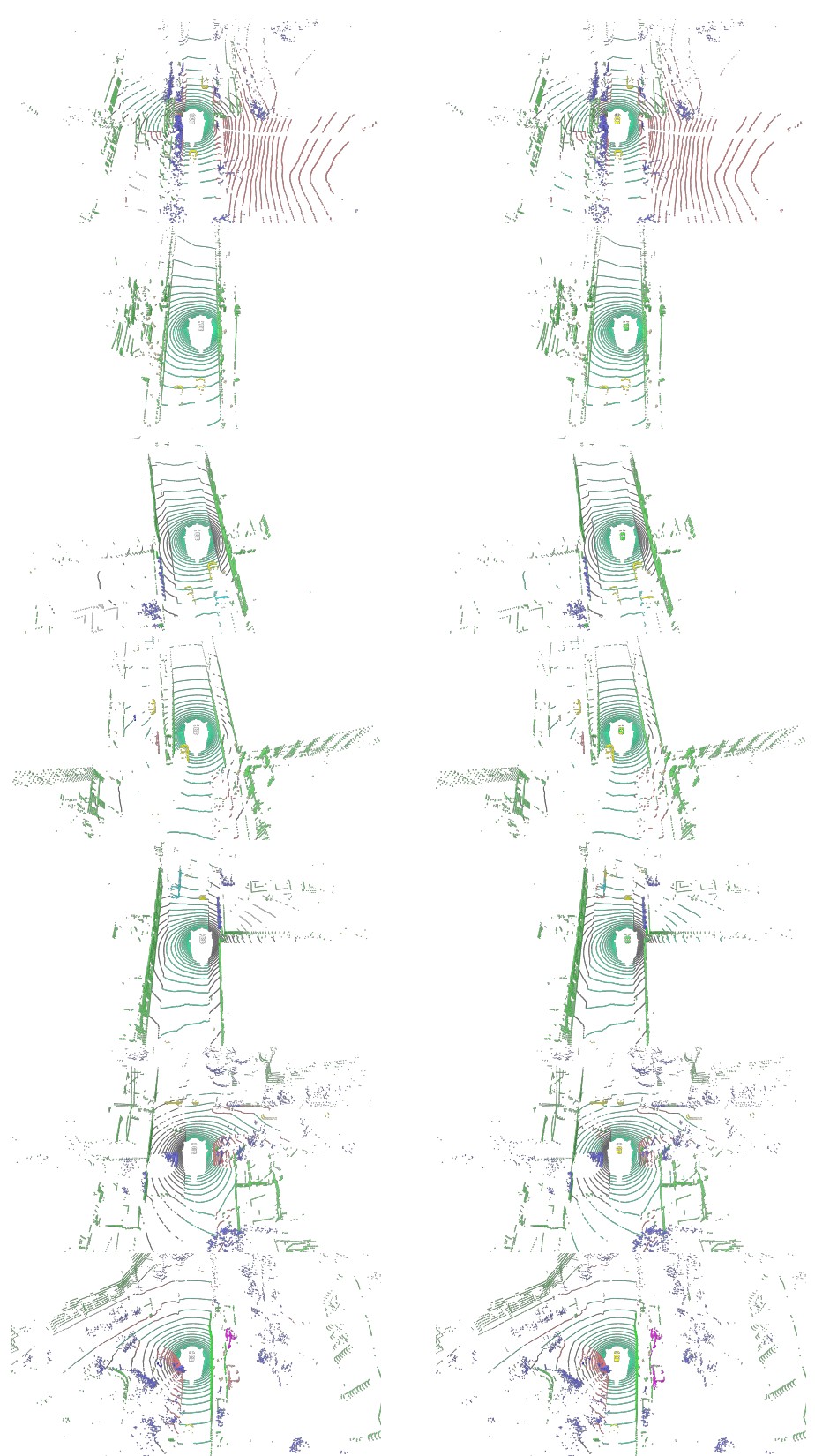

Figure 9: Visualization of ground-truth label (left) and our LiDAR segmentation results (right) on nuScenes Fong et al. (2022).

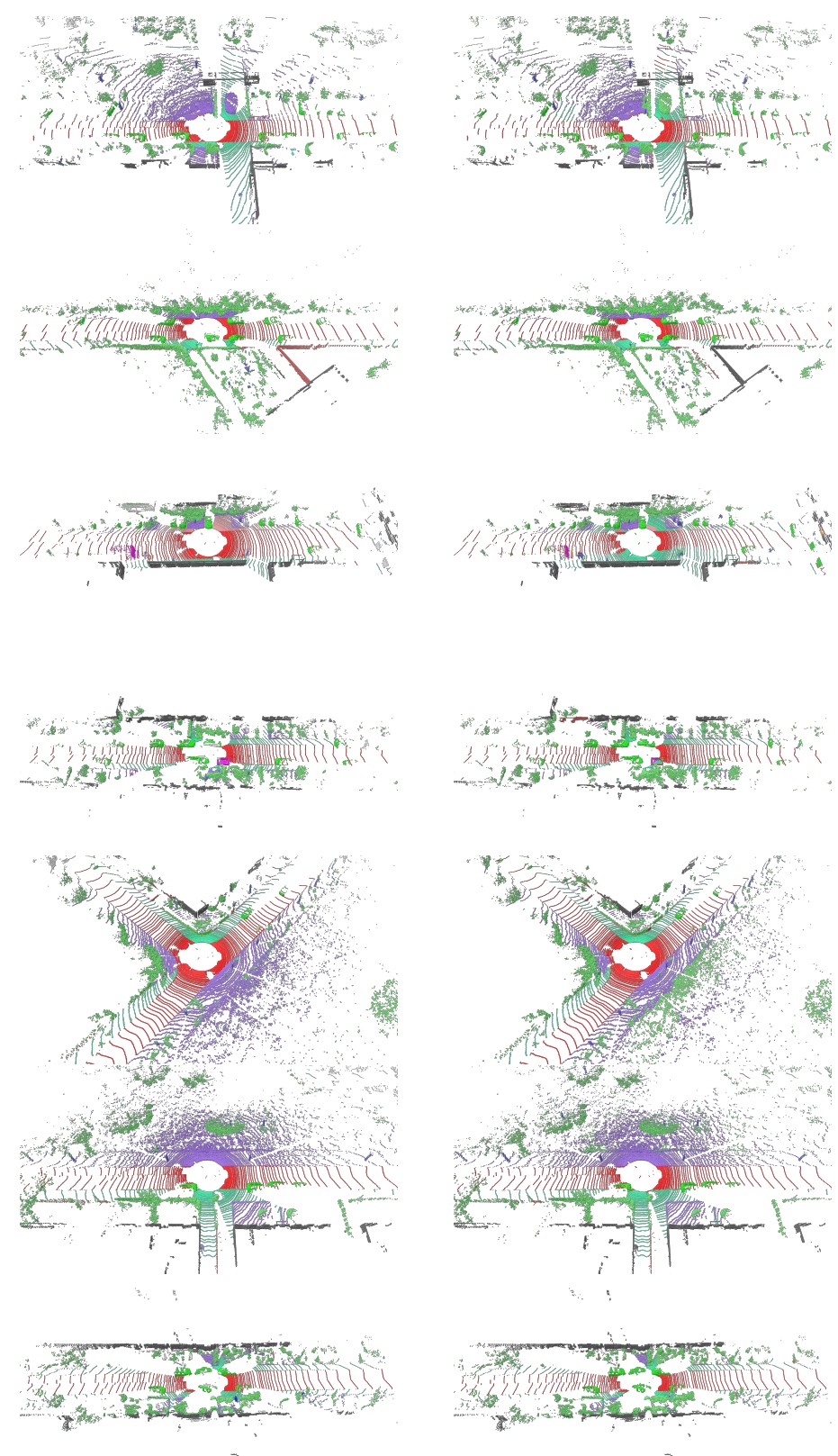

Figure 10: Visualization of ground-truth label (left) and our LiDAR segmentation results (right) on SemanticKITTI Behley et al. (2019).

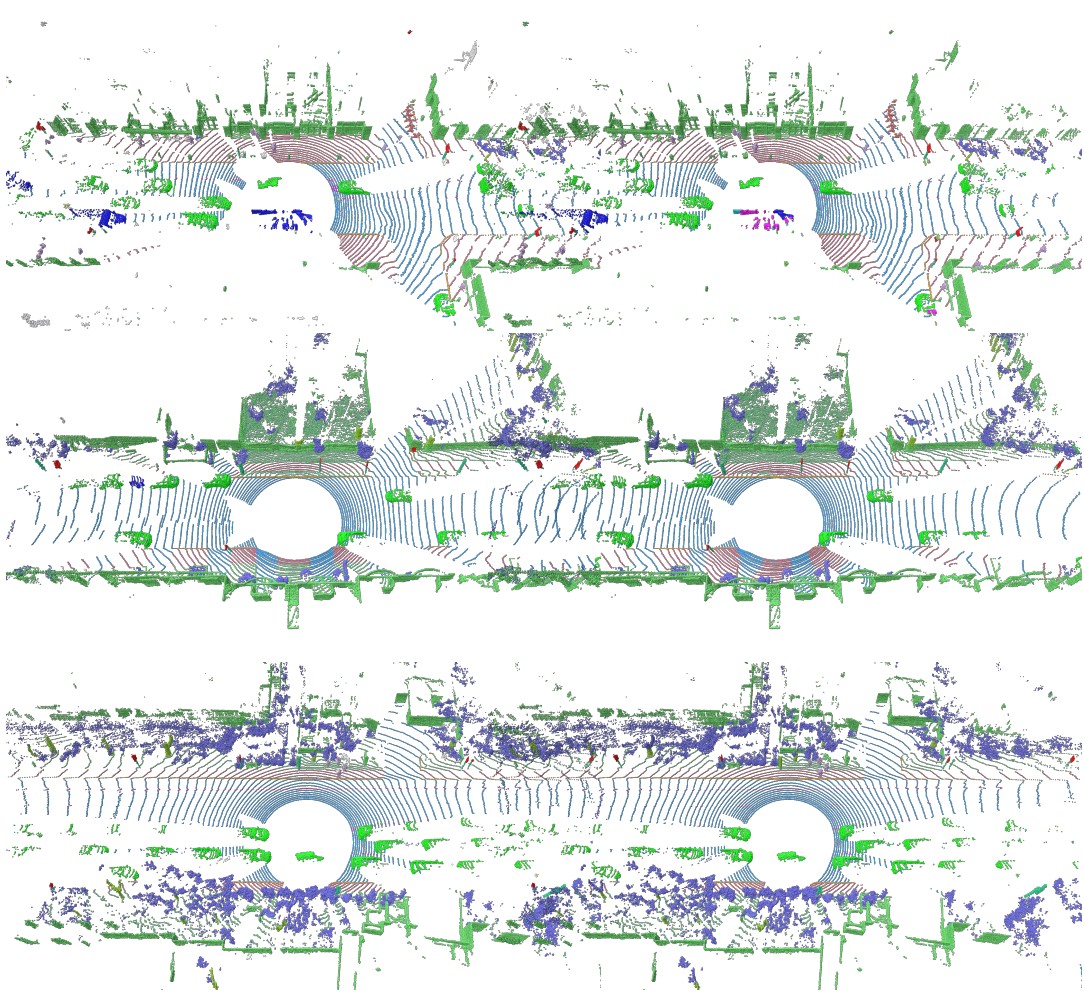

Figure 11: Visualization of ground-truth label (left) and our LiDAR segmentation results (right) on Waymo Sun et al. (2020).

