# OpenReview forum: "Surface Representation in LiDAR Scenes"
_ICLR.cc/2024/Conference — Submitted to ICLR 2024_

### Official Review · Reviewer_i3Qi · 2023-10-29

**Soundness:** 3 good
**Presentation:** 3 good
**Contribution:** 3 good
**Rating:** 6
**Confidence:** 3

**Summary:**

In this work, the authors propose a representation dedication for LiDAR point clouds. It particularly extends the work RepSurf for synthetic point clouds to the case of LiDAR point clouds. To tackle the challenge of sparse, uneven, and large-scale LiDAR point clouds, the authors use a point sliding module to jitter the centroid of the triangle. An additional strategy is to gradually downsample the point clouds and extract the features for aggregation. This work demonstrates its performance in point cloud segmentation and shows some advantages.

**Strengths:**

This paper is well-written and easy to follow. The idea to extend RepSurf to the LiDAR point cloud case is also interesting. Experimentation seems to support the capability of the proposal for the application of segmentation.

**Weaknesses:**

There are quite a few concerns about this work:

1. This work only applies the proposed method for feature extraction to segmentation. Can it be generalized to other types of point cloud problems, for instance, low-level point cloud processing problems, e.g., denoising, upsampling, etc.?

2. A follow-up question is, since this method works well for sparse LiDAR point clouds, can it be applied back to the simpler case which is the dense point clouds or synthetic point clouds? Will the performance also be improved?

3. The Point Sliding Module generates some jittering to the centroid for augmentation. This process has certain randomness because the coefficients are drawn from uniform or Gaussian distribution. I wonder how stable this method would be in the end, because there is a certain randomness in the process.

4. The authors use downsampling with FPS to obtain the RealSurf feature. However, this is also performed in PointNet++. Thus it would be better if the authors could further clarify the way they do the downsampling differs from that is proposed in PointNet++.

5. How this this work compare to other types of advanced feature extractors? For example, those based on transformer modules, e.g., Point transformer and Voxel transformer.

6. How is the computational complexity of this proposal compared to PointNet++ and other competing methods?

**Questions:**

Please refer to the concerns/questions/comments mentioned in the weaknesses part.

---

> ### Author Response · Authors · 2023-11-23
> **Response to Reviewer i3Qi**
>
> We sincerely thank the reviewer for providing constructive feedback and helping us shaping the current draft. We have included a supplement for revision, and we provide responses to specific questions:
>
> **Can it be generalized to other types of point cloud problems?**
>
> We believe so, and that would be a great direction for future work. As the reviewer points out, our method serves as surface representation for LiDAR point clouds, and thus can be applied to other point cloud problems, as a way to address unevenness and low point density. For example, for any problem that involves upsampling, we can adopt point sliding module (proposed in Sec 3.3) to densify LiDAR point clouds.
>
> **Can it be applied back to the simpler case which is the dense point clouds or synthetic point clouds?**
>
> We can apply this technique to the simpler case, however, our goal is to address issues inherent to the processing of noisy and uneven point clouds (e.g., LiDAR point clouds). Therefore, RealSurf will not be beneficial in these settings, although performance should not degrade.
>
> **How stable this method would be in the end for randomness**
>
> Experimental results may vary slightly due to random sampling. In our experiments we empirically observed a small oscillation of $0.5$, regardless of different benchmarks.
>
> **The difference between our downsampling with the FPS in PointNet++**
>
> As shown in Sec. 3.3, the intent of adopting FPS is not to propose a novel downsampling strategy, but to construct regular triangles on uneven point clouds.
> That is, we utilize an off-the-shelf method to obtain relatively uniform point clouds for the surface feature computation on LiDAR point clouds.
>
> **Compare to other advanced feature extractors.**
>
> We conducted some additional experiments on the feature extractors such as PointNeXt [1] and PointMLP [2] on SemanticKITTI val set (mIoU, \%) below:
>
>
> |                 | PointNeXt [1]   | PointMLP [2]    | PointNet++ ($\times$1) |
> |-----------------|-----------------|-----------------|:-----------------------|
> | w/o RealSurf    | 51.5            | 52.4            | 51.3                   |
> | **w/ RealSurf** | **60.9** (+9.4) | **61.2** (+8.8) | **62.3** (+11.0)       |
>
> **Computational complexity.**
>
> Similar to RepSurf [3], our RealSurf introduces minor extra computation compared to PointNet++. We also compare with other competitive methods on SemanticKITTI as follows:
>
>
> |                          | mIoU | \#Params (M) | Inference (s/sample) | Memory (GB) |
> |--------------------------|------|--------------|:---------------------|-------------|
> | Cylinder3D [4]              | 65.9 | 55.8         | 0.13                 | 1.62        |
> | LiDARMultiNet [5]           | 69.1 | 29.2         | 0.41                 | 6.35        |
> | **RealSurf ($\times$2)** | 68.5 | 3.9          | 1.16                 | 2.06        |
> | **RealSurf ($\times$4)** | 70.2 | 15.3         | 1.72                 | 4.14        |
>
>
> **A Sincere Apology:**
>
> Again, We sincerely thank the reviewer for initiating these insightful discussions to further strengthen our submission. We apologize for the delayed response due to the CVPR deadline. We welcome any further comments or suggestions and commit to addressing them thoroughly in the camera-ready version of our paper.
>
> **Additional Reference:**
>
> [1] Guocheng Qian, Yuchen Li, Houwen Peng, Jinjie Mai, Hasan Abed Al Kader Hammoud, Mohamed Elhoseiny, and Bernard Ghanem. Pointnext: Revisiting pointnet++ with improved training and
> scaling strategies. arXiv preprint arXiv:2206.04670, 2022
>
> [2] Xu Ma, Can Qin, Haoxuan You, Haoxi Ran, and Yun Fu. Rethinking network design and local
> geometry in point cloud: A simple residual mlp framework. arXiv preprint arXiv:2202.07123,
> 2022
>
> [3] Haoxi Ran, Jun Liu, and Chengjie Wang. Surface representation for point clouds. In Proceedings of the IEEE/CVF Conference on Computer Vision and Pattern Recognition, pp. 18942–18952, 2022
>
> [4] Xinge Zhu, Hui Zhou, Tai Wang, Fangzhou Hong, Yuexin Ma, Wei Li, Hongsheng Li, and Dahua
> Lin. Cylindrical and asymmetrical 3d convolution networks for lidar segmentation. In Proceed-
> ings of the IEEE/CVF conference on computer vision and pattern recognition, pp. 9939–9948,
> 2021
>
> [5] Dongqiangzi Ye, Zixiang Zhou, Weijia Chen, Yufei Xie, Yu Wang, Panqu Wang, and Hassan
> Foroosh. Lidarmultinet: Towards a unified multi-task network for lidar perception. arXiv preprint
> arXiv:2209.09385, 2022

---

### Official Review · Reviewer_TgQF · 2023-10-30

**Soundness:** 2 fair
**Presentation:** 2 fair
**Contribution:** 2 fair
**Rating:** 5
**Confidence:** 4

**Summary:**

This paper presents RealSurf, a framework designed to process point clouds in challenging environments like autonomous driving. The paper identifies challenges in applying surface representations to real scans and offers solutions, including the Point Sliding Module for geometric feature computation and a LiDAR-based surfel reconstruction process that reduces unevenness. When evaluated on benchmarks like nuScenes, SemanticKITTI, and Waymo, RealSurf outperforms its competitors and establishes its efficiency. This success highlights the potential of point-based methods in LiDAR segmentation.

**Strengths:**

1. The presented work offers a new framework designed for processing LiDAR point clouds.
2. Using a basic PointNet++ structure, this method has demonstrated top-tier performance across multiple demanding datasets, such as SemanticKITTI, nuScenes, and Waymo.
3. Their innovative solutions are versatile and can be integrated into any point-based network.

**Weaknesses:**

1. The paper would benefit from further refinement in its writing. Some sections might appear a bit complex and could be clarified for easier comprehension.
2. Many parts of the writing are not professional, such as many lines in the text that have only one word, which should be avoided as much as possible.
3. It might be beneficial for readers if the motivation behind the paper were more prominently emphasized. Upon careful reading, I found it a bit subtle to identify the core motivation. Was it primarily about the 'point sliding module'? If so, the contribution of the article appears to be somewhat incremental. I hope the authors can highlight their contributions.
4. The experimental results are good, but the comparison might not seem very fair. If I'm not mistaken, the entire framework is just PointNet++ with the addition of tangent plane information to its input. Why is there such a significant improvement compared to PointNet++ as shown in Table 1 or Table 2? Does the comparison in Table 1 or 2 involve an unfair comparison? How would it fare against other methods if various tricks were also applied? In Table 2, compared to the previous PointNet++, there's an improvement of 50 points. It seems that some of the latest tricks were used. It's hard to believe that there would be such a significant improvement by just adding some extra information. If there are any unfair comparisons, it would be better to unify some settings for a fair comparison and demonstrate it on top-performing (SOTA) methods. After all, the method introduced in the paper is framework-agnostic and is a general method for point clouds.

**Questions:**

All the questions I want to ask are in the 'weakness' section.

---

> ### Author Response · Authors · 2023-11-23
> **Response to Reviewer TgQF**
>
> We sincerely thank the reviewer for providing constructive feedback and helping us shaping the current draft. We have included a supplement for revision, and we provide responses to specific questions:
>
> **Further refinement in its writing and many parts of the writing are not professional:**
>
> We appreciate the constructive feedback and have updated our main text to fix them. We will thoroughly proof-read the paper for the camera-ready version.
>
> **Highlight their contributions.**
>
> We summarize our contributions in Sec. 3.3 as follows:
> 1) We introduce a novel augmentation method (point sliding module) that does not require much computation and is specifically designed for surface representation for outdoor scenes.
> 2) We conduct downsampling before surfel construction to avoid distorted triangles for the computation of surface features.
>
> Simultaneously, in Sec.3.4:
>
> 3) We deliberately design a novel baseline with the simple backbone PointNet++ to bridge the gap between point-based methods and voxel-based methods for LiDAR semantic segmentation.
>
> **The comparison might not seem very fair.**
>
> As shown in Table 5, the improvements come from our three proposed designs: point sliding module for surface-level augmentation,
> applying downsampling for regular triangle construction (Sec. 3.3), and
> implementation optimization for point-based methods to compete with voxel-based methods (Sec. 3.4).
>
> Note that, these implementation findings are *all* inspired by the common techniques from voxel-based pipelines: [1] firstly propose ground-truth augmentation for class balancing, and [2] adopt Lovász-Softmax loss [3] for positive/negative sample balancing in LiDAR segmentation. Accordingly, we design a point-based baseline to be adequately competitive with popular voxel-based methods. However, these findings cannot lead point-based methods much better than voxel-based methods without our designs in Sec. 3.3.
>
> Additionally, as shown in Sec. 4.1, we have pointed out that our method is trained and evaluated in the same conditions as other methods, i.e. we use the same training data and validation protocols. Thus, any improvements come directly from our contributions, leading to our reported state-of-the-art results.
>
> **A Sincere Apology:**
>
> Again, We sincerely thank the reviewer for initiating these insightful discussions to further strengthen our submission. We apologize for the delayed response due to the CVPR deadline. We welcome any further comments or suggestions and commit to addressing them thoroughly in the camera-ready version of our paper.
>
> **Additional Reference:**
>
> [1] Yan Yan, Yuxing Mao, and Bo Li. Second: Sparsely embedded convolutional detection. Sensors,
> 18(10):3337, 2018
>
> [2] Xinge Zhu, Hui Zhou, Tai Wang, Fangzhou Hong, Yuexin Ma, Wei Li, Hongsheng Li, and Dahua
> Lin. Cylindrical and asymmetrical 3d convolution networks for lidar segmentation. In Proceed-
> ings of the IEEE/CVF conference on computer vision and pattern recognition, pp. 9939–9948,
> 2021
>
> [3] Maxim Berman, Amal Rannen Triki, and Matthew B Blaschko. The lov ́asz-softmax loss: A tractable
> surrogate for the optimization of the intersection-over-union measure in neural networks. In Pro-
> ceedings of the IEEE conference on computer vision and pattern recognition, pp. 4413–4421,
> 2018

---

### Official Review · Reviewer_7faG · 2023-11-01

**Soundness:** 3 good
**Presentation:** 3 good
**Contribution:** 2 fair
**Rating:** 5
**Confidence:** 4

**Summary:**

The paper proposes to generate surface elements (surfels) from LiDAR point clouds. Different from the previous approach that only works on synthetic data, the proposed method, RealSurf, is able to process the point clouds in real data. To solve this problem, the proposed method includes a Point Sliding Module that jitters points within the reconstructed surfels, and a LiDAR-based surfel reconstruction process leveraging attenuating unevenness. The proposed method is evaluated on three popular outdoor benchmarks, such as nuScenes, SemanticKITTI, and Waymo, and archives state-of-the-art results even compared to voxel-based and RGB+point methods.

**Strengths:**

1. The method is evaluated extensively on three popular outdoor benchmarks and achieves state-of-the-art performance.
2. Thorough ablation studies are conducted to show the effectiveness of each proposed module.

**Weaknesses:**

1. Naive baseline: Occupancy Network + Marching Cubes?
2. How many points are left after a different number of downsampling?
3. Outdoor lidar point cloud is more sparse, I am wondering how the proposed method works on indoor dense point cloud with more occlusions in real scans?
4. Point Sliding is a data augmentation method that jitters the points so as to make the proposed method more robust to noises. In the ablation study (Figure 7), the authors show 0.5 performs best with Guassian noise on SemanticKITTI. I am wondering, is this hyper parameter consistent across different benchmarks?
5. Is the method limited to pointnet++?
6. I am not sure jitter points count as a novel contribution for dealing with noisy real scans. It is common to apply this technique when training networks on noisy point clouds, e.g, see MinkovskiEngine dataloader.

**Questions:**

The voxel-based method also shows some advantages, do you consider a hybrid network that consumes both points and voxels? such as Point-Voxel CNN for Efficient 3D Deep Learning.

---

> ### Author Response · Authors · 2023-11-23
> **Response to Reviewer 7faG**
>
> We sincerely thank the reviewer for providing constructive feedback and helping us shaping the current draft. We have included a supplement for revision, and we provide responses to specific questions:
>
> **Naive baseline: Occupancy Network + Marching Cubes?**
>
> Occupancy network [1], a method developed on synthetic data and based on marching cubes algorithm [2], is not common in LiDAR point clouds due to the limited efficiency and noises on larger-scale point clouds. Thus, there is very few previous work for reference. However, since occupancy network is heavily based on volumes, we consider its alternative voxel-based baselines (e.g., MinkowskiNet [3], PolarNet [4]) in LiDAR scenes. We have updated our main text to compare our proposed method to MinkowskiNet [3] and PolarNet [4] on SemanticKITTI. Ours outperforms MinkowskiNet (mIoU: 53.2) implemented by [5] and PolarNet (mIoU: 69.4) by 17.5 and 10.7 mIoU, respectively. This comparison further demonstrates the superiority of our approach as a novel baseline.
>
> **How many points are left after a different number of downsampling?**
>
> We utilize *all* 3D points of a LiDAR point cloud as our input without any preprocessing like voxelization. For a point cloud of $N$ points, after each downsampling operation, the resultant number of points decreases by a factor of 4. Specifically, for the $i$-th downsampling, the output number of points becomes $N/4^i$.
>
> **How the proposed method works on indoor dense point cloud with more occlusions in real scans?**
>
> In our paper, we did not apply our method to indoor scenes since our proposed modules in Sec. 3.3 are specifically designed for outdoor scenes:
> 1) Our motivation of applying downsampling before computing surface representation comes from the extreme unevenness of LiDAR point clouds, which is not obvious in dense point clouds like indoor scans (they are more uniformly sampled).
> 2) Since these point clouds are usually dense, indoor scans do not need to be further densified through our proposed densification module.
>
> However, we agree that this analysis deserves a further investigation in the camera-ready version of the paper.
>
> **Is this hyper parameter consistent across different benchmarks?**
>
> We have empirically determined an optimal set of hyperparameters and use the same ones on all considered benchmarks. This indicates that our method is not overly sensitive to the choice of hyperparameters. Thus, we choose the factor of 0.5 as our default setting across different benchmarks. We will include a discussion about this topic in the camera-ready version, and provide further analysis.
>
> **A Sincere Apology:**
>
> Again, We sincerely thank the reviewer for initiating these insightful discussions to further strengthen our submission. We apologize for the delayed response due to the CVPR deadline. We welcome any further comments or suggestions and commit to addressing them thoroughly in the camera-ready version of our paper.
>
> **Additional Reference:**
>
> [1] Lars Mescheder, Michael Oechsle, Michael Niemeyer, Sebastian Nowozin, and Andreas Geiger. Occupancy networks: Learning 3d reconstruction in function space. In Proceedings of the IEEE/CVF
> conference on computer vision and pattern recognition, pp. 4460–4470, 2019
>
> [2] William E Lorensen and Harvey E Cline. Marching cubes: A high resolution 3d surface construction
> algorithm. In Seminal graphics: pioneering efforts that shaped the field, pp. 347–353. 1998
>
> [3] Christopher Choy, JunYoung Gwak, and Silvio Savarese. 4d spatio-temporal convnets: Minkowski
> convolutional neural networks. In Proceedings of the IEEE/CVF Conference on Computer Vision
> and Pattern Recognition, pp. 3075–3084, 2019
>
> [4] Yang Zhang, Zixiang Zhou, Philip David, Xiangyu Yue, Zerong Xi, Boqing Gong, and Hassan
> Foroosh. Polarnet: An improved grid representation for online lidar point clouds semantic seg-
> mentation. In Proceedings of the IEEE/CVF Conference on Computer Vision and Pattern Recog-
> nition, pp. 9601–9610, 2020
>
> [5] Alexey Nekrasov, Jonas Schult, Or Litany, Bastian Leibe, and Francis Engelmann. Mix3d: Out-of-context data augmentation for 3d scenes. In 2021 International Conference on 3D Vision (3DV), pp. 116–125. IEEE, 2021

---

### Official Review · Reviewer_AzRp · 2023-11-06

**Soundness:** 2 fair
**Presentation:** 3 good
**Contribution:** 2 fair
**Rating:** 5
**Confidence:** 4

**Summary:**

This paper proposes a surface representation approach for the lidar point cloud semantic segmentation.
Based on the RepSurf, the authors proposed RealSurf by mainly introducing a point sliding module and FPS sampling to address challenges by sparsity, density variation, and large scale as analyzed. Evaluation results on three benchmarks show the advantages of the proposed method.

**Strengths:**

- The paper is well-written and easy to follow.
- The motivation of the method is practical and reasonable for processing LiDAR point cloud.
- The experiments on there datasets show the advatanges of RealRep.

**Weaknesses:**

- The paper misses comparison with recent point cloud works such as [1].
- In my understanding, the main purpose of the introduced strategies seems to make the real data uniform for Repsurf and make the model more robust to real noisy data by jitter augmentation. The technical contributions are somewhat limited considering the RepSurf.


[1] Spherical Transformer for LiDAR-based 3D Recognition. CVPR 2023

**Questions:**

- The efforts of proposed strategies such as downsampling seem to trim the data more suitable for Repsurf.  How about the effects if applying the same strategies to synthetic data?
- It is a little confusing why the number of points doesn't increase after the densification process as illustrated in Fig. 2.

---

> ### Author Response · Authors · 2023-11-23
> **Response to Reviewer AzRp**
>
> We sincerely thank the reviewer for providing constructive feedback and helping us shaping the current draft. We have included a supplement for revision, and we provide responses to specific questions:
>
> **The paper misses comparison with recent point cloud works such as [1]**
>
> We didn't notice this very recent paper and have included it in the revision. Indeed, SphereFormer [1] achieves strong performance on some datasets. However, it leverages complicated Transformer architectures which have much more parameters (32.3M vs ours: 15.3M). Due to time limit and an intensive CVPR submission during this rebuttal period, we cannot incorporate these techniques of [1] into our architecture and conduct a fair comparison. We have included [1] in the related work section.
>
> **The technical contributions are somewhat limited considering the RepSurf**
>
> In this paper, we aim to explore the feasibility of adopting surface representation in the context of LiDAR scenes. As mentioned in Figure1 and Section 3, a direct application of surface representation (i.e., RepSurf) to LiDAR point clouds simply fails, given how much performance degrades in this challenging setting. Though simple, our contributions enable the use of surface representation in LiDAR scenes, in a way that previous methods are incapable of. Hence, we believe that our contributions are relevant and provide useful insights to the scientific community.
>
> **How about the effects if applying the same proposed strategies such as downsampling to synthetic data?**
>
> It will not be as effective, since synthetic point clouds are already uniformly sampled in a manner similar to what FPS aims to achieve. However, performing downsampling like FPS on LiDAR point clouds will be quite different, as it can greatly mitigate the effects of the unevenness for the process of surface feature computation.
>
> **Why the number of points doesn't increase after the densification process as illustrated in Fig. 2.**
>
> Our ultimate goal is to tackle the sparsity issues by upsampling LiDAR point clouds. We densify the point clouds by interpolating points, which leads to unaffordable memory consumption. Therefore, we abandon the original points and only keep the interpolated ones. This operation does not sacrifice performance while maintaining efficiency. Conceptually, it is a densification-then-downsampling process, and that is why we call it ``densification''. We will clarify this point and revise the names in the final version.
>
> **A Sincere Apology:**
>
> Again, We sincerely thank the reviewer for initiating these insightful discussions to further strengthen our submission. We apologize for the delayed response due to the CVPR deadline. We welcome any further comments or suggestions and commit to addressing them thoroughly in the camera-ready version of our paper.
>
> **Additional Reference:**
>
> [1] Xin Lai, Yukang Chen, Fanbin Lu, Jianhui Liu, and Jiaya Jia. Spherical transformer for lidar-based 3d recognition. In Proceedings of the IEEE/CVF Conference on Computer Vision and Pattern Recognition, pp. 17545–17555, 2023

---

### Meta-Review · Area_Chair_VrHF · 2023-12-06

**Metareview:**

This paper presents a surface representation for LiDAR point clouds that deals with sparseness, unevenness, and large-scale under environments like autonomous driving.  To this end, a point sliding module for jitter augmentation and down-sampling with FPS for feature extraction are proposed. The proposed method is evaluated on three popular outdoor benchmarks to demonstrate SOTA performance.  The paper mostly is well written and the proposed method is reasonable and practical for processing LiDAR point clouds.  On the other hand, missing a recent work presented at CVPR23, applicability of the proposed method to synthetic data or indoor scene where occlusions happen, and novelty of point jittering were raised as main concerns by the reviewers.  Although the authors’ rebuttal resolved some of the concerns, some concerns still remain.  First, time limitation due to CVPR submission is irrelevant and should not be counted.  Issues regarding occlusions and point jittering are not fully addressed.  Moreover, since the authors claim that the propose method is developed to deal with sparsity and unevenness of point clouds, required is detailed investigation on how sparse or uneven should be expected for the proposed method to effectively work.  In this sense, evaluation under controlled sparsity/unevenness is mandatory to support the claimed key contributions.  These were discussed and acknowledged in the post-rebuttal discussion.  Substantial revision followed by at least one more round of peer review is required for publication.  The paper cannot be accepted, accordingly.

**Justification For Why Not Higher Score:**

N/A

**Justification For Why Not Lower Score:**

N/A

---

### Decision · Program_Chairs · 2024-01-16

Reject